# Socio-Economic Inequalities in the Double Burden of Malnutrition among under-Five Children: Evidence from 10 Selected Sub-Saharan African Countries

**DOI:** 10.3390/ijerph20085489

**Published:** 2023-04-12

**Authors:** Olufunke A. Alaba, Plaxcedes Chiwire, Aggrey Siya, Oluremi A. Saliu, Karen Nhakaniso, Emmanuella Nzeribe, Denis Okova, Akim Tafadzwa Lukwa

**Affiliations:** 1Health Economics Unit, School of Public Health and Family Medicine, Faculty of Health Sciences, University of Cape Town, Anzio Road, Observatory, Cape Town 7925, South Africa; 2Department of Health Services Research, CAPHRI Care and Public Health Research Institute, Maastricht University, 6200 MD Maastricht, The Netherlands; 3Western Cape Department: Health, Western Cape Province, P.O. Box 2060, Cape Town 8000, South Africa; 4College of Veterinary Medicine, Animal Resources and Biosecurity, Makerere University, Kampala P.O. Box 7062, Uganda; 5Department of Environmental Health Science, Faculty of Health Sciences, National Open University of Nigeria, Abuja 901101, Nigeria; 6Business School, University of the People, 595 E Colorado Blvd Suite 623, Pasadena, CA 91101, USA; 7Faculty of Pharmacy and Pharmaceutical Sciences, Kwame Nkrumah University of Science and Technology, Kumasi GPS AK-448-4944, Ghana; 8DSI-NRF Centre of Excellence in Epidemiological Modelling and Analysis (SACEMA), Stellenbosch University, Private Bag X1, Matieland, Stellenbosch 7602, South Africa

**Keywords:** under 5 years, double burden of malnutrition, SES inequality, SSA, concentration indices

## Abstract

Background: Africa is unlikely to end hunger and all forms of malnutrition by 2030 due to public health problems such as the double burden of malnutrition (DBM). Thus, the aim of this study is to determine the prevalence of DBM and degree of socio-economic inequality in double burden of malnutrition among children under 5 years in sub-Saharan Africa. Methods: This study used multi-country data collected by the Demographic and Health Surveys (DHS) Program. Data for this analysis were drawn from the DHS women’s questionnaire focusing on children under 5 years. The outcome variable for this study was the double burden of malnutrition (DBM). This variable was computed from four indicators: stunting, wasting, underweight and overweight. Inequalities in DBM among children under 5 years were measured using concentration indices (CI). Results: The total number of children included in this analysis was 55,285. DBM was highest in Burundi (26.74%) and lowest in Senegal (8.80%). The computed adjusted Erreygers Concentration Indices showed pro-poor socio-economic child health inequalities relative to the double burden of malnutrition. The DBM pro-poor inequalities were most intense in Zimbabwe (−0.0294) and least intense in Burundi (−0.2206). Conclusions: This study has shown that across SSA, among under-five children, the poor suffer more from the DBM relative to the wealthy. If we are not to leave any child behind, we must address these socio-economic inequalities in sub-Saharan Africa.

## 1. Introduction

According to the World Food Programme (WFP), malnutrition is “a state in which the physical function of an individual is impaired to the point where he or she can no longer maintain adequate bodily performance process such as growth, pregnancy, lactation, physical work and resisting and recovering from disease” [1]. Malnutrition can also be termed as an abuse of food or bad nutrition, such as over-nutrition and under-nutrition. Hunger and protein-energy malnutrition (PEM) have led to high mortality rates in children and mothers, contributing to poor growth and a rise in opportunistic infections and comprised brain development [2]. On the other hand, overweight and obesity develop because of over-nutrition and have led to an increase in major chronic diseases, which can be life-threatening sometimes. Additionally, micronutrient deficiency has also become rampant in children, which is the major cause of the slow development in children at their developmental stage [3]. The most common micronutrient deficiencies clinically shown to have affected over two billion children under under-five include iron, calcium, zinc, and vitamin A [4]. In the past, many fortification programs have been implemented, drastically improving the situation; however, some developing countries still suffer a significant rate of morbidity [5]. Malnutrition presents a significant threat to humans and has been a public health problem that has resulted in about 45% of childhood deaths [6].

Stunting has been cited to be related to poverty and poor diet [7]. In China, while the prevalence of undernutrition among children has reduced [8], a worrisome new trend is emerging: the co-existence of under-nutrition and over-nutrition [9], and the rising prevalence of overweight and obesity especially in urban areas and affluent rural areas [10]. Despite experiencing economic growth in recent times, India still reports high rates of child malnutrition with 35%, 33% and 17% of children under 5 years being stunted, underweight and wasted, respectively [11]. As of 2019, the number of malnourished people was reported to be around 27% and 29% of the total population for Eastern and Middle sub-regions of Sub-Saharan Africa [12]. These rates were projected to grow, with Sub-Saharan Africa unlikely to achieve the Sustainable Development Goals of ending hunger and all forms of malnutrition by 2030 [12]. The Food and Agriculture Organization (FAO) reported 161 million children under 5 to be stunted as of 2020 [13]. At the same time, 3.4 million people die each year due to being overweight and obese [14]. Between 2000 and 2017, the number of children under 5 presenting with stunting rose from 50.6 to 58.7 million, and the number of overweight children under 5 rose from 6.6 million in 2000 to 9.7 million in 2017 [15]. In 2020, which happens to be the pandemic year recorded, over 140 million under-five children were stunted, 59 million were wasted, and 85 million were moderately or severely underweight [13]. Today, Africa bears a double burden of undernutrition and over-nutrition [16].

The coexistence of two contrasting extremes on the malnutrition spectrum, such as undernutrition and overweight, is known as the double burden of malnutrition (DBM) [17]. DBM can present at an individual, household, or population level. An individual can present both stature and obesity, or different members within a household could be underweight whilst the other is overweight. Many low-income and middle-income countries (LMICs) grapple with challenges associated with DBM [18]. A 2018 World Health Organization (WHO) series found that DBM poses a significant public health threat, as high levels of undernutrition and overweight are rising in LMIC [19]. Sub-Saharan Africa has been characterized by a high prevalence of undernutrition and increasing obesity, which is a typical example of the double burden of malnutrition [6]. Considering the latter at the current status, there is a high chance that Africa may not achieve Sustainable Development Goal 2 (SDG2) by 2030. There are a lot of factors that may have increased the under-nutrition in Africa than in other continents. One may be improper environmental sanitation, which depicts the growth of micronutrients in the environment [13,20]

The DBM is intricately bound to the socio-economic conditions present in sub-Saharan Africa. Socio-economic factors include occupation, education, income, wealth, and where an individual resides [21]. Inequalities arising from socio-economic factors will exacerbate the major causes of malnutrition in children under 5 [22]. Inadequate nutritional intake, insufficient care and household discriminatory food distribution are some of the causes of malnutrition in LMIC [23]. Nutrition in children is heavily reliant on socio-economic factors [24]. Studies have found that children from higher socio-economic status are more likely to have healthier food habits than those from low socio-economic background who display unhealthier food habits that will contribute to malnutrition and have dietary profiles consistent with nutritional guidelines [12,22,23,24,25,26,27,28,29,30,31,32].

The underlying causes of DBM vary by sub-region in sub-Saharan Africa. For example, one study found that cultural perceptions such as a heavier body size in females may signify wealth, good, stable marital home, and exceptional achievement [33]. However, these perceptions differ across sub-regions as some regions expect women to work hard and have increased physical activity. Another study attributed obesity and increased weight to the rise in consumption of cheap, processed food at the expense of fresh, non-processed foods of subsistence farming [34]. The rise in the commercialization of food production is correlated to the decrease in subsistence farming. This has led to household diets with low nutritional value, high sugar, high fat, and energy-dense food that leads to obesity [35]. This article uses a population-based study on ten African countries (Burundi, Ethiopia, Guinea, Malawi, Mali, Senegal, Sierra Leone, South Africa, Zambia, and Zimbabwe) to understand the socio-economic inequalities in the double burden malnutrition among under-5 children in the continent. Thus, the aim of this study is to determine the prevalence of DBM and degree of socio-economic inequality in double burden of malnutrition among children under 5 years in sub-Saharan Africa.

## 2. Materials and Methods

### 2.1. Data Source

This study used multi-country data collected by the Demographic and Health Surveys (DHS) Program. The countries included in this study were: Zimbabwe (2015), Malawi (2015), Burundi (2016), South Africa (2016), Guinea (2018), Mali (2018), Zambia (2018), Sierra Leone (2019), Senegal (2019), and Ethiopia (2019). Data for this analysis were drawn from the DHS women’s questionnaire focusing on children under 5 years (under 60 months). DHS employs a two-stage sampling method. Sample sizes for children under 5 included in our analysis are as follows: Zimbabwe (*n* = 5253), Malawi (5384), Burundi (6096), South Africa (1460), Guinea (3582), Mali (8908), Zambia (9100), Sierra Leone (4540), Senegal (5682), and Ethiopia (5280). These 10 countries were chosen based on the availability of recent data for secondary analysis.

### 2.2. Study Variables

#### 2.2.1. Stunting

Children whose height-for-age Z-score was below minus three standard deviations (−3 SD) were considered stunted, and those above minus three standard deviations (−3 SD) were considered not stunted. Stunting was coded as binary variable assigned values of zero and one. Those that were not stunted above −3 SD were coded as “0”, and those that were stunted below −3 SD were coded as “1”.

#### 2.2.2. Wasting

Children whose Z-score was below minus three standard deviations (−3 SD) from the median of the reference population were considered thin (wasted), and those above a Z-score above −3 SD were considered not wasted. Wasting was coded as binary variable assigned values of zero and one. Those that were not wasted above −3 SD were coded as “0” and those that were wasted below −3 SD were coded as “1”.

#### 2.2.3. Underweight

Children whose weight-for-age Z-score was below minus three standard deviations (−3 SD) from the median of the reference population were classified as underweight, and those that were above −3 SD considered not underweight. Underweight was coded as binary variable assigned values of zero and one. Those that were not underweight above −3 SD were coded as “0” and those that were underweight below −3 SD were coded as “1”.

#### 2.2.4. Overweight

Children whose weight-for-height Z-score was more than 2 standard deviations (+2 SD) above the median of the reference population were considered overweight. Overweight was coded as binary variable assigned values of zero and one. Those that were not overweight below +2 SD were coded as “0” and those that were overweight above +2 SD were coded as “1”.

#### 2.2.5. Double Burden of Malnutrition (DBM)

The outcome variable for this study is the double burden of malnutrition (DBM). This variable was computed from 4 indicators: stunting, wasting, underweight and overweight. The first step was calculating the row total of stunted, wasted, and underweight children. Then children who had a row total greater than 1 and were overweight were defined as having experienced a double burden of malnutrition. The outcome variable, double burden of malnutrition (DBM), was then recorded as a binary variable where a value of “1” was assigned if DBM was present, and a value of “0” was given if DBM was absent.

#### 2.2.6. Socio-Economic Status (SES)

Socio-economic status was depicted by the household wealth index, which measures a household’s cumulative standard of living, such as ownership of select assets, type of housing, sanitation services, and type of water access, among others, using Principal Component Analysis (PCA) [32]. The household wealth index is considered a more reliable measure of wealth compared to income and consumption because it reflects a household’s long-term standard of living, and this makes it possible to identify problems particular to the poor members of society, such as unequal access to health care and unequal access to recommended nutrition [33]. For this study, wealth was grouped into 5 quintiles—poorest (Q1), poorer (Q2), middle (Q3), richer (Q4) and richest (Q5).

### 2.3. Statistical Analysis

#### Data Analysis

The study analyzed the data using STATA 17.1 statistical software. Univariate and bivariate analyses were performed to describe the sample and patterns of DBM. Inequalities in DBM among children under 5 years were measured using concentration indices (CI). The concentration index approach is a standard measure of assessing health inequalities. The indices and curves investigate whether health inequalities exist in one group. However, they do not estimate the magnitude of health inequalities [36]. This paper used the Erreygers normalized concentration indices [37] to measure the socio-economic inequalities among children in DBM: wasting, stunting, underweight and overweight. Among many of the indices that could have been used, we opted to adopt the Normalized Erreygers Indices as they have been corrected for bound issues; hence, they give more robust standard errors. The concentration index ranges from −1 to +1 and estimates the extent to which a health outcome (DBM) is concentrated among the rich or the poor. A negative concentration index value denotes a health outcome (DBM) concentrated among the poor. In contrast, a positive value implies that a health outcome (DBM) is concentrated among the rich [36]. A concentration index of zero implies that there is no socio-economic-related inequality, and a large absolute value of the concentration index depicts a greater concentration of inequality [36,38].

The concentration index can be computed by making use of the ‘covariance’ as shown below:(1)$$CI=\frac{2}{\hat{y}}COV\left(y_{i},R_{i}\right)$$
where: *y_i_* is the health variable;

ŷ is the mean of *y_i_*;

*R_i_* is the fractional rank of the *i*th individual;

*COV* denotes the covariance.

## 3. Results

### 3.1. Demographic Characteristics

After data cleaning, the total number of children included in this analysis was 55,285. Across the 10 countries, the majority of the children resided in the rural areas (Burundi (84.25%), Ethiopia (76.97%), Guinea (71.44%), Malawi (83.67%), Mali (74.15%), Senegal (70.89%), Sierra Leone (70.55%), Zambia (70.47%), and Zimbabwe (63.14)) except for South Africa (48.63) (Table 1). Across all countries, a higher prevalence of DBM was reported in rural areas compared to urban areas (Table 2).

### 3.2. Prevalence of Wasting, Stunting, Underweight and Overweight

The total average prevalence of stunting across the 10 countries was 11.6% (Figure 1A). Of 10 countries, 3 were above the total average: Ethiopia (12.2%), Guinea (13.7%) and Burundi (24.8%), and Burundi had the highest stunting prevalence of (24.8%) (Figure 1A). Zambia had the same stunting prevalence as the total average of 11.6%, while the remaining 6 countries had stunting prevalence below the total average: Malawi (10.5%), Sierra Leone (10.3%), Mali (10%), South Africa (9.9%), Zimbabwe (8.1%) and Senegal, which had the lowest stunting prevalence of 4.8% (Figure 1A).

Meanwhile, for wasting, the prevalence was highest in Guinea (3.6%) and lowest in South Africa (0.5%) (Figure 1B). Zambia (1.5%), Mali (2.6%) and Guinea (3.6%) reported the wasting prevalence values that were higher than the total average of 1.4% (Figure 1B). In contrast, Senegal (1.3%), Ethiopia (1.2%), Zimbabwe (1.1%), Sierra Leone (1.0%), Burundi (0.9%), Malawi (0.6%) and South Africa (0.5%) had wasting prevalence values that were below the total average (Figure 1B). Burundi (8.3%) had the highest underweight prevalence again, while South Africa (1.0%) had the lowest underweight prevalence (Figure 1C). Guinea (5.3%), Mali (5.3%), Ethiopia (5.8%) and Burundi (8.3%) had underweight prevalence values which were above the total average (Figure 1C). Meanwhile, Sierra Leone (3.4%), Senegal (3.0%), Malawi (2.5%), Zambia (2.4%), Zimbabwe (1.4%) and South Africa (1.0%) had underweight prevalence values that were below the total underweight average for the 10 countries (3.8%) (Figure 1C). Additionally, the overweight prevalence was highest in South Africa (13.8%) and lowest in Burundi (1.4%) (Figure 1D). Sierra Leone (5.0%), Zambia (5.4%), Guinea (6.0%), Zimbabwe (6.0%) and South Africa (13.8%) had overweight prevalence values above the overweight total average (4.9%) (Figure 1D).

### 3.3. Socio-Economic Inequalities

Across all countries, stunting disproportionately affected the poorest, as stunting was more prevalent in the poorest quintile (Q1) (Table 3). The concentration indices for stunting were all negative and statistically significant at a 95% confidence interval in all countries, ranging from −0.21 in Burundi to −0.03 in Zimbabwe (Table 3).

Burundi, Ethiopia, Guinea, Mali, Senegal, Sierra Leone, and Zimbabwe all reported pro-poor wasting child health inequalities (Table 4). The latter concentration indices were statistically significant at a 95% confidence interval. While Malawi, South Africa and Zambia reported pro-rich inequalities, the concentration indices were not statistically significant at a 95% confidence interval (Table 4).

Underweight children were also more prevalent in the poorest quintile (Q1) for most of the countries except for Zimbabwe (31.43%; Q2) and South Africa (35.71; Q3): Burundi (38.84%), Ethiopia (51.52%), Guinea (30.69%), Malawi (34.11%), Mali (28.94%), Senegal (52.50%), Sierra Leone (33.33%), and Zambia (38.57%) (Table 5). All the underweight concentration indices were statistically significant at a 95% confidence interval across countries except for South Africa which had pro-rich inequalities; otherwise, all other countries reported negative indices, indicating pro-poor underweight inequalities (Table 5).

Conversely, overweight disproportionately affected children from the rich households in many of the countries (Burundi, Ethiopia, Malawi, Mali, Senegal, Sierra Leone, Zambia and Zimbabwe); however, only Burundi, Mali, Senegal and Zimbabwe had statistically significant concentration indices at a 95% confidence interval (Table 6). While Guinea and South Africa reported pro-poor inequalities, the concentration indices were not statistically significant at a 95% confidence interval (Table 6).

DBM was highest in Burundi (27.4%), followed by Guinea (21.5%), Zambia (16.7%), Ethiopia (16.6%), South Africa (15.5%), Malawi (15.2%), Sierra Leone (14.3%), Mali (14.1%), Zimbabwe (13.6%) and lowest in Senegal (8.6%) (Figure 2). Across all countries, DBM was most prevalent among children in the poorest quintile (Q1) except in Zimbabwe, where DBM was most prevalent among children from the richer quintile (Q4) (Table 7). However, the adjusted Erreygers concentration indices were negative, showing pro-poor DBM inequalities among children across all countries (Table 7).

DBM across all countries reported pro-poor inequalities as all the concentration indices were negative and statistically significant at a 95% confidence interval (Table 8). The computed adjusted Erreygers Concentration Indices showed pro-poor socio-economic child health inequalities relative to the double burden of malnutrition. All the concentration indices were negative across all countries and were statistically significant at a 95% confidence interval (Table 8). The DBM pro-poor inequalities were more intense in Zimbabwe (−0.0294) and least intense in Burundi (−0.2206) (Table 8). Meanwhile, between Zimbabwe and Burundi, the intensity of the child health pro-poor inequalities was as follows: Sierra Leone (−0.0356), Zambia (−0.0460), Malawi (−0.0536), Senegal (−0.0597), South Africa (−0.0652), Mali (−0.0689), Guinea (−0.0765) and Ethiopia (−0.1161) (Table 8).

## 4. Discussion

SDG Target 2.2 aims to “End all forms of malnutrition, including achieving, by 2020, the internationally agreed targets on stunting and wasting in children under 5 years of age”. In this regard, our study contributes several ways to the debate on DBM among children in African countries. First, our results provide evidence on individual nutritional status (underweight, wasting, stunting, and overweight) prevalence of children under 5 years of age at the national level and explain the existence of socio-economic inequalities. The quality of evidence for our approach is supported by representative DHS data from 10 African countries. The completeness of the dataset for analysis, as this study has drawn insights from the most recent dataset from 2015 to 2019, suggests that the geographic and social differences in DBM of under-five children in Africa and the extent of economic inequality can be fully understood.

Sub-Saharan Africa has been cited to be characterized by the double burden of malnutrition (DBM) and high levels of undernutrition as well as a growing burden of overweight/obesity and diet-related non-communicable diseases (NCDs) [39]. Recent research shows that despite a high prevalence of hunger and malnutrition, overweight and obesity epidemics are increasing in Africa [40]. This is still the case, as our study findings showed a significantly high prevalence of DBM, with Senegal reporting the least DBM prevalence of about 9%. In comparison, Burundi had the highest DBM prevalence of about 27%. A recent study raised a concern, citing the possibility that most countries will not meet the global nutrition targets by 2030 [39] and Africa is unlikely to reach the Sustainable Development Goals and end hunger and all forms of malnutrition by 2030. The current study findings seem to show the concern raised in earlier papers becoming a sad reality as this study reported a significantly high prevalence of stunting (Burundi; 24%), underweight (Burundi; 8%) and overweight (South Africa; 13%). Earlier research reported malnutrition commonly observed in developed and affluent communities, but as early as 1996, it was noticed in low-to-middle-income countries (LMICs) [41,42,43].

A recent study reported the prevalence of overweight and obesity among under-five children in South Africa to be almost double that of Malawi [44]. Our results also showed similar findings: South Africa had the highest overweight prevalence of 13% compared to Senegal, which had about 2%. However, it had the 5th highest DBM prevalence of 16%, with Burundi with the highest DBM prevalence of 27%. The reported that the high DBM prevalence of Burundi could be attributed to earlier trends of stunting and underweight among children, which have shown little to no changes since the 1980s [43].

Contrary to what was observed in previous studies that reported relatively socio-economic well-off groups at a greater risk for the double burden of malnutrition [44,45,46], our study showed that DBM was more prevalent among the children from the poorest households (Q1). This may be because of a shift in NCDs’ epidemiology, as they were earlier perceived as diseases for developed countries but are currently more prevalent in developing countries. Furthermore, a DBM is a global problem. It has been argued that it occurs when the prevalence of overweight and obesity in LMICs is increasing rapidly, while at the same time, the prevalence of malnutrition in these countries is declining slowly [47]. This was true for our study, as Burundi had the highest prevalence of stunting (24%) and underweight (8%). As a result, it had the highest DBM prevalence (27%).

Obesity in children under 5 years of age is still overlooked in the current literature. Our study provided evidence of the increasing burden of obesity in this age group, which was found primarily in households of high socio-economic status. Previous studies reported similar findings arguing that wealthier groups pose strong risk factors for the double burden of malnutrition as well as community-level poverty [48,49,50]. Considering that findings from this study showed that DBM is intertwined in underweight, stunting, wasting and overweight, addressing the social inequalities that share the double burden of child malnutrition in the African region therefore requires strategies that address why certain sub-populations are more exposed to these nutritional problems to avoid strategies that solve one nutritional problem and exacerbate another. 

There is a need to increase the engagement of various stakeholders to mitigate the double burden of malnutrition in sub-Saharan Africa. The active collaboration and participation of representatives from local and international non-governmental organizations, major corporations, and government institutions across various sectors such as agriculture, finance, environment, education, communications, health care and nutrition will possibly stimulate dialogues around this menace and proffer solutions and recommendations. Furthermore, progress toward ending hunger and malnutrition by 2030 requires intensified efforts to reduce undernutrition and focused action on reducing obesity and diet-related non-communicable diseases. Key strengths of this study lie in it being a compilation of representative and generalizable DHS datasets from 10 countries. These are typically high-quality, highly responsive datasets from DHS surveys conducted using robust methodologies using well-documented data sources. These DHS surveys are conducted using standardized survey modules and implementations that allow comparisons between countries. However, these are cross-cutting datasets, which limited our ability to assign causality.

## 5. Conclusions

In summary, there is a shift in nutrition in Africa, with the increasing prevalence of overweight and obesity among children under five, making optimal child nutrition a key factor in achieving global health goals. The inequality of DBM was consistently pro-poor socio-economic across the ten SSA countries, such that the lower socio-economic groups were more likely to be experiencing DBM and bear a higher burden of the problem than the higher socio-economic groups. In addition, DBM was pro-poor although some of the nutritional indicators were pro-rich. Therefore, the double burden of malnutrition in low- and middle-income countries poses a major global public health problem that could hinder the achievement of the SDGs if not properly addressed. Furthermore, this study has shown the existence of pro-poor inequalities relative to the double burden of malnutrition among under-five children. Therefore, if we are not to leave any child behind, we must address these socio-economic inequalities in sub-Saharan Africa.

## Figures and Tables

**Figure 1 ijerph-20-05489-f001:**
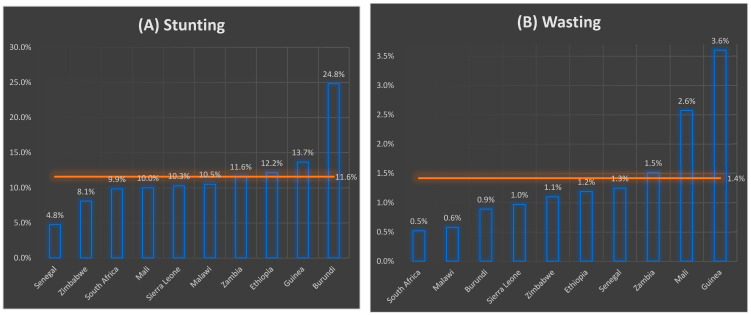
(**A**–**D**) Prevalence of wasting, stunting, underweight and overweight by country.

**Figure 2 ijerph-20-05489-f002:**
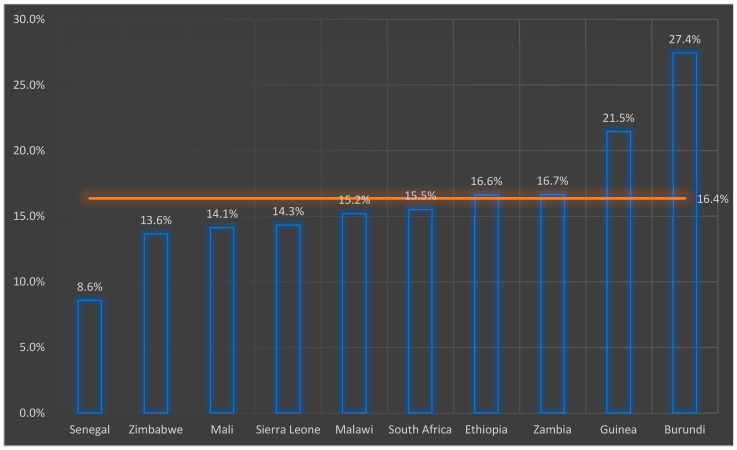
Double burden of malnutrition by country.

**Table 1 ijerph-20-05489-t001:** Demographic characteristics by country (N = 55,285).

Country	Year	Total Sample N %	Mean Age ± S.D. [Confidence Interval]	Socio-Economic Status		Residence Status	
				Q1_poorest_ N %	Q5_richest_ N %	Rural N %	Urban N %
** *Burundi* **	2016	6096 (11.03)	28.92 + 17.58 [28.48 29.36]	1235 (20.26)	1227 (20.13)	5136 (84.25)	960 (15.75)
** *Ethiopia* **	2019	5280 (9.55)	28.91 + 17.24 [28.44 29.38]	1788 (33.86)	1153 (21.84)	4064 (76.97)	1216 (23.03)
** *Guinea* **	2018	3582 (6.48)	27.59 + 17.17 [27.03 28.15]	882 (24.62)	532 (14.85)	2559 (71.44)	1023 (28.56)
** *Malawi* **	2016	5384 (9.74)	28.97 + 17.16 [28.51 29.43]	1179 (21.90)	949 (17.63)	4505 (83.67)	879 (16.33)
** *Mali* **	2018	8908 (16.11)	28.18 + 17.20 [27.82 28.54]	1773 (19.90)	1592 (17.87)	6605 (74.15)	2303 (25.85)
** *Senegal* **	2019	5682 (10.28)	28.06 + 17.32 [27.61 28.51]	1724 (30.34)	561 (9.87)	4028 (70.89)	1654 (29.11)
** *Sierra Leone* **	2019	4540 (8.21)	27.03 +17.38 [26.53 27.54]	1207 (26.59)	523 (11.52)	3203 (70.55)	1337 (29.45)
** *South Africa* **	2016	1460 (2.64)	29.02 + 17.47 [28.13 29.92]	342 (23.42)	156 (10.68)	710 (48.63)	750 (51.37)
** *Zambia* **	2018	9100 (16.46)	28.42 + 17.34 [28.06 28.77]	2596 (28.53)	1159 (12.74)	6413 (70.47)	2687 (29.53)
** *Zimbabwe* **	2015	5253 (9.50)	28.37 + 17.43 [27.90 28.84]	1109 (21.11)	1036 (19.72)	3317 (63.14)	1936 (36.86)

**Table 2 ijerph-20-05489-t002:** Estimated prevalence of children who experience DBM by residence status.

Country	Rural %	SE	Confidence Interval	Urban %	SE	Confidence Interval
** *Burundi* **	29.32	0.0064	[0.28 0.31]	12.92	0.0108	[0.11 0.15]
** *Ethiopia* **	18.85	0.0061	[0.18 0.20]	10.12	0.0087	[0.09 0.12]
** *Guinea* **	22.82	0.0082	[0.21 0.25]	16.32	0.0116	[0.14 0.19]
** *Malawi* **	15.52	0.0054	[0.15 0.17]	11.95	0.0109	[0.10 0.14]
** *Mali* **	15.44	0.0045	[0.16 0.17]	12.81	0.0070	[0.12 0.14]
** *Senegal* **	9.51	0.0046	[0.09 0.11]	7.07	0.0063	[0.06 0.08]
** *Sierra Leone* **	15.80	0.0064	[0.15 0.17]	14.14	0.0095	[0.12 0.16]
** *South Africa* **	18.45	0.0146	[0.16 0.22]	14.13	0.0127	[0.12 0.17]
** *Zambia* **	17.07	0.0047	[0.16 0.18]	16.11	0.0071	[0.15 0.18]
** *Zimbabwe* **	14.08	0.0060	[0.13 0.15]	13.02	0.0076	[0.12 0.15]

**Table 3 ijerph-20-05489-t003:** Estimated prevalence of stunting among children by socio-economic status (SES) quintile.

Country	SES Quintiles					Absolute Difference Q1–Q5	Relative Difference Q1/Q5	Erreygers Norm. CI	SE
	Poorest Q1	Poorer Q2	Middle Q3	Richer Q4	Richest Q5				
** *Burundi* **	30.25	25.41	22.44	15.54	6.35	23.90	4.76	−0.2181 ***	0.0125
** *Ethiopia* **	44.41	21.00	15.11	11.63	7.85	36.56	5.66	−0.1008 ***	0.0132
** *Guinea* **	30.00	26.52	23.26	12.17	8.04	21.96	3.73	−0.0712 ***	0.0136
** *Malawi* **	31.01	25.50	17.43	14.86	11.19	19.82	2.77	−0.0652 ***	0.0106
** *Mali* **	26.05	24.65	23.26	17.44	8.60	17.45	3.03	−0.0643 ***	0.0080
** *Senegal* **	47.99	28.19	11.41	9.40	3.02	44.97	15.89	−0.0516 ***	0.0082
** *Sierra Leone* **	34.06	22.34	21.91	12.36	9.33	24.73	3.65	−0.0424 ***	0.0131
** *South Africa* **	40.21	23.71	20.62	11.34	4.12	36.09	9.76	−0.0684 ***	0.0190
** *Zambia* **	36.16	25.07	16.78	14.37	7.62	28.54	4.75	−0.0531 ***	0.0087
** *Zimbabwe* **	27.69	20.26	15.90	24.87	11.28	16.41	2.45	−0.0383 ***	0.0089

*** indicate statistical significance at 99.9%.

**Table 4 ijerph-20-05489-t004:** Estimated prevalence of wasting among children by socio-economic status (SES) quintile.

Country	SES Quintiles					Absolute Difference Q1–Q5	Relative Difference Q1/Q5	Erreygers norm. CI	SE
	Poorest Q1	Poorer Q2	Middle Q3	Richer Q4	Richest Q5				
** *Burundi* **	26.42	39.62	16.98	13.21	3.77	22.65	7.01	−0.0099 ***	0.0025
** *Ethiopia* **	66.67	15.56	4.44	5.56	7.78	58.89	8.57	−0.0258 ***	0.0054
** *Guinea* **	26.67	20.83	19.17	18.33	15.00	11.67	1.78	−0.0008	0.0080
** *Malawi* **	11.76	23.53	23.53	17.65	23.53	−11.77	0.50	0.0033	0.0024
** *Mali* **	28.93	21.90	19.01	14.88	15.29	13.64	1.89	−0.0138 **	0.0047
** *Senegal* **	59.52	10.71	15.48	8.33	5.95	53.57	10.00	−0.0161 ***	0.0051
** *Sierra Leone* **	34.62	25.00	15.38	15.38	9.62	25.00	3.60	−0.0050 **	0.0041
** *South Africa* **	20.00	20.00	30.00	20.00	10.00	10.00	2.00	0.0036	0.0064
** *Zambia* **	29.27	22.76	14.63	17.89	15.45	13.82	1.89	0.0015	0.0037
** *Zimbabwe* **	29.31	24.14	13.79	22.41	10.34	18.97	2.83	−0.0078 *	0.0040

***, ** and * indicate statistical significance at 99.9%, 99% and 95%, respectively.

**Table 5 ijerph-20-05489-t005:** Estimated prevalence of underweight among children by socio-economic status (SES) quintile.

Country	SES Quintiles					Absolute Difference Q1–Q5	Relative Difference Q1/Q5	Erreygers Norm. CI	SE
	Poorest Q1	Poorer Q2	Middle Q3	Richer Q4	Richest Q5				
** *Burundi* **	38.84	25.83	19.01	12.81	3.51	35.33	11.07	−0.1058 ***	0.0078
** *Ethiopia* **	51.52	21.88	10.80	8.31	7.48	44.04	6.89	−0.0741 ***	0.0115
** *Guinea* **	30.69	24.34	19.58	16.40	8.99	21.70	3.41	−0.0229 **	0.0094
** *Malawi* **	34.11	27.13	15.50	14.73	8.53	25.58	4.00	−0.0203 ***	0.0052
** *Mali* **	28.94	26.13	24.62	14.47	5.83	23.11	4.96	−0.0488 ***	0.0061
** *Senegal* **	52.50	24.00	11.00	10.50	2.00	50.50	26.25	−0.0387 ***	0.0075
** *Sierra Leone* **	33.33	17.61	19.50	16.35	13.21	20.12	2.52	−0.0044	0.0080
** *South Africa* **	21.43	21.43	35.71	14.29	7.14	14.29	3.00	0.0016	0.0069
** *Zambia* **	38.57	27.62	12.38	13.33	8.10	30.47	4.76	−0.0141 ***	0.0038
** *Zimbabwe* **	28.57	31.43	15.71	14.29	10.00	18.57	2.86	−0.0123 ***	0.0038

*** and ** indicate statistical significance at 99.9% and 99% respectively.

**Table 6 ijerph-20-05489-t006:** Estimated prevalence of overweight among children by socio-economic status (SES) quintile.

Country	SES Quintiles					Absolute Difference Q1–Q5	Relative Difference Q1/Q5	Erreygers Norm. CI	SE
	Poorest Q1	Poorer Q2	Middle Q3	Richer Q4	Richest Q5				
** *Burundi* **	14.58	20.83	14.58	19.79	30.21	−15.63	0.48	0.0078 *	0.0040
** *Ethiopia* **	24.73	20.43	8.60	15.05	31.18	−6.45	0.79	0.0095	0.0050
** *Guinea* **	26.29	20.10	24.23	11.86	17.53	8.76	1.50	−0.0013	0.0104
** *Malawi* **	17.75	23.81	20.78	18.61	19.05	−1.30	0.93	0.0075	0.0066
** *Mali* **	18.43	16.13	18.43	20.28	26.73	−8.30	0.69	0.0103 *	0.0052
** *Senegal* **	22.35	16.47	15.29	25.88	20.00	2.35	1.12	0.0140 ***	0.0045
** *Sierra Leone* **	29.90	16.67	20.10	14.71	18.63	11.27	1.60	0.0080	0.0098
** *South Africa* **	25.35	26.76	21.13	20.42	6.34	19.01	4.00	−0.0057	0.0229
** *Zambia* **	28.63	21.48	20.17	15.62	14.10	14.53	2.03	0.0046	0.0062
** *Zimbabwe* **	19.03	15.16	14.84	26.77	24.19	−5.16	0.79	0.0197 **	0.0084

***, ** and * indicate statistical significance at 99.9%, 99% and 95%, respectively.

**Table 7 ijerph-20-05489-t007:** Estimated prevalence of DBM among children by socio-economic status (SES) quintile.

Country	SES Quintiles					Absolute Difference Q1–Q5	Relative Difference Q1/Q5	Erreygers Norm. CI	SE
	Poorest Q1	Poorer Q2	Middle Q3	Richer Q4	Richest Q5				
** *Burundi* **	29.51	24.72	22.09	15.89	7.79	21.72	3.79	−0.2206 ***	0.0125
** *Ethiopia* **	44.43	20.81	13.05	11.25	10.46	33.97	4.25	−0.1161 ***	0.0146
** *Guinea* **	28.76	24.63	21.97	13.05	11.58	17.18	2.48	−0.0765 ***	0.0170
** *Malawi* **	26.49	24.88	19.15	15.80	13.68	12.81	1.94	−0.0536 ***	0.0116
** *Mali* **	25.32	23.19	22.36	17.03	12.09	13.23	2.09	−0.0689 ***	0.0102
** *Senegal* **	45.20	23.80	13.00	11.80	6.20	39.00	7.29	−0.0597 ***	0.0110
** *Sierra Leone* **	32.52	21.01	21.15	13.24	12.09	20.43	2.69	−0.0356 **	0.0154
** *South Africa* **	29.54	25.32	21.94	17.30	5.91	23.63	5.00	−0.0652 ***	0.0214
** *Zambia* **	33.90	23.95	17.87	14.33	9.95	23.95	3.41	−0.0460 ***	0.0099
** *Zimbabwe* **	24.20	18.92	15.02	25.03	16.83	7.37	1.44	−0.0294 **	0.0114

*** and ** indicate statistical significance at 99.9% and 99% respectively.

**Table 8 ijerph-20-05489-t008:** Erreygers Normalized Concentration Indices for DBM by country.

Country	Erreygers Norm. CI	Robust Standard Error	*p*-Value
** *Burundi* **	−0.2206 ^1^	0.0125	0.00
** *Ethiopia* **	−0.1161 ^2^	0.0146	0.00
** *Guinea* **	−0.0765 ^3^	0.0170	0.00
** *Malawi* **	−0.0536 ^4^	0.0116	0.00
** *Mali* **	−0.0689 ^5^	0.0102	0.00
** *Senegal* **	−0.0597 ^6^	0.0101	0.00
** *Sierra Leone* **	−0.0356 ^7^	0.0154	0.00
** *South Africa* **	−0.0652 ^8^	0.0236	0.00
** *Zambia* **	−0.0460 ^9^	0.0100	0.00
** *Zimbabwe* **	−0.0294 ^10^	0.0114	0.01

^1^ Burundi (Note: Standard error adjusted for 554 clusters in primary sampling unit); ^2^ Ethiopia (Note: Standard error adjusted for 305 clusters in primary sampling unit); ^3^ Guinea (Note: Standard error adjusted for 400 clusters in primary sampling unit); ^4^ Malawi (Note: Standard error adjusted for 850 clusters in primary sampling unit); ^5^ Mali (Note: Standard error adjusted for 345 clusters in primary sampling unit); ^6^ Senegal (Note: Standard error adjusted for 214 clusters in primary sampling unit); ^7^ Sierra Leone (Note: Standard error adjusted for 572 clusters in primary sampling unit); ^8^ South Africa (Note: Standard error adjusted for 546 clusters in primary sampling unit); ^9^ Zambia (Note: Standard error adjusted for 545 clusters in primary sampling unit; ^10^ Zimbabwe (Note: Standard error adjusted for 399 clusters in primary sampling unit).

## Data Availability

All data sets are publicly available on the Demographic Health Survey website at: https://dhsprogram.com/what-we-do/survey/survey-display-406.cfm (accessed on 10 January 2023) and can be accessed upon request from the Demographic Health Survey team.

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
