# Peer review of "Socio-Economic Inequalities in the Double Burden of Malnutrition among under-Five Children: Evidence from 10 Selected Sub-Saharan African Countries"

_ijerph, 2023, doi:10.3390/ijerph20085489_

Round 1
Reviewer 1 Report
This is an interesting and potentially very informative article. The authors have conducted a secondary analysis of data from the Demographic and Health Survey conducted in 10 African countries. The aim is to investigate the Double Burden of Malnutrition (DBM) in these countries and also consider how DBM relates to household wealth.
The main issue with the present manuscript is the multiple errors in the results section. It appears the authors have used a template and/or citation software that has produced multiple error messages. Lines 200-202 seem to be guidance from the template that has not been revised or removed.
It also looks as though there may be errors in the figures and tables. Figure 1, for example, does not include all countries and the quadrants are not labelled a,b,c,d although the figure details include these letters (with the exception of 'a').
It wasn't clear what data from the Demographic Health Survey (or elsewhere) was used to estimate stunting, wasting, underweight and overweight. More detail is needed about how the data were collected, cutpoints used etc.
The introduction should have a stronger focus more on DBM, with a clear definition provided in the first paragraph.
The abstract exceeds IJERPH's 200 word limit.
There is reference to over-nutrition, which should probably refer to overweight/obesity. Overweight and malnutrition can co-exist - overweight doesn't indicate too much nutrition.
Line 79 - should this be 'comprised brain development' rather than just 'brain development'?
Line 147-148 - should these be 'under 60 months' rather than 'under 59 months'?
Author Response
Dear International Journal of Environmental Research Public Health (IJERPH) Editorial Office,
Thank you for the opportunity to resubmit a revised draft of our manuscript titled “Socio-economic inequalities in the double burden of malnutrition among under-five children: evidence from 10 selected sub-Saharan African countries” to International Journal of Environmental Research Public Health. We appreciate the time and effort that you and the reviewers have dedicated to providing valuable feedback on our manuscript. We have reviewed the comments and incorporated changes in the manuscript to reflect suggestions provided by the reviewers. We have highlighted the changes within the manuscript and we have also provided responses to some of the comments. Below are detailed responses to each of your valuable comments.
REVIWER 1 COMMENTS AND RESPONSES
Comment 1 Reviewer 1
This is an interesting and potentially very informative article. The authors have conducted a secondary analysis of data from the Demographic and Health Survey conducted in 10 African countries. The aim is to investigate the Double Burden of Malnutrition (DBM) in these countries and also consider how
DBM relates to household wealth.
Response to Comment 1 from Reviewer 1
We are pleased with the above comment and we thank you for your input which has been relevant to improving the quality of our manuscript.
Comment 2 Reviewer 1
The main issue with the present manuscript is the multiple errors in the results section. It appears the authors have used a template and/or citation software that has produced multiple error messages. Lines 200-202 seem to be guidance from the template that has not been revised or removed.
Response to Comment 2 from Reviewer 1
We had been using cross reference in citing tables and figures this has been resolved in the updated manuscript. All changes are highlighted in yellow.
Comment 3 Reviewer 1
It also looks as though there may be errors in the figures and tables. Figure 1, for example, does not include all countries and the quadrants are not labelled a,b,c,d although the figure details include these letters (with the exception of 'a').
Response to Comment 3 from Reviewer 1
Thank you this has been resolved, the other countries were not picking up as the graphs had been compressed when formatting (Lines 242-243);
|
|
|
|
|
Comment 4 Reviewer 1
The introduction should have a stronger focus more on DBM, with a clear definition provided in the first paragraph.
Response to Comment 4 from Reviewer 1
Thank you this has been resolved, line 86 to line 124;
“The coexistence of two contrasting extremes on the malnutrition spectrum, such as undernutrition and overweight, is known as the double burden of malnutrition (DBM) [17]. DBM can present at an individual, household, or population level. An individual can present both stature and obesity, or different members within a household could be underweight whilst the other is overweight. Many low-income and middle-income countries (LMICs) grapple with challenges associated with DBM [18]. A 2018 World Health Organization (WHO) series found that DBM poses a significant public health threat as high levels of undernutrition and overweight are rising in LMIC [19]. Sub-Saharan Africa has been characterized by a high prevalence of undernutrition and increasing obesity, a typical example of the double burden of malnutrition [6] Considering the latter at the current status, there is a high chance that Africa may not achieve sustainable development goal 2 (SDG - 2) by 2030. There are a lot of factors that may have increased the under-nutrition in Africa than in other continents. One may be improper environmental sanitation, which depicts the growth of micronutrients in the environment [13,20]
The DBM is intricately bound to the socio-economic conditions present in Sub-Saharan Africa. Socio-economic factors include occupation, education, income, wealth, and where an individual resides [21]. Inequalities arising from socio-economic factors will exacerbate the major causes of malnutrition in children under-5 [22]. Inadequate nutritional intake, insufficient care and household discriminatory food distribution are some of the causes of malnutrition in LMIC [23]. Nutrition in children is heavily reliant on socio-economic factors [24]. Studies have found that children from higher socio-economic status are more likely to have healthier food habits than those from low socio-economic background who display unhealthier food habits that will contribute to malnutrition and have dietary profiles consistent with nutritional guidelines [12,22–32].
The underlying causes of DBM vary by sub-region in Sub-Saharan Africa. For example, one study found that cultural perceptions such as a heavier body size in females may signify wealth, good, stable marital home, and exceptional achievement [33]. However, these perceptions differ across sub-regions as some regions expect women to work hard and have increased physical activity. Another study attributed obesity and increased weight to the rise in consumption of cheap, processed food at the expense of fresh, non-processed foods of subsistence farming [34]. The rise in the commercialization of food production is correlated to the decrease in subsistence farming. This has led to household diets with low nutritional value, high sugar, high fat, and energy-dense food that leads to obesity [35]. This article uses a population-based study on ten African countries (Burundi, Ethiopia, Guinea, Malawi, Mali, Senegal, Sierra Leone, South Africa, Zambia, and Zimbabwe) to understand the socio-economic inequalities in the double burden malnutrition among inder-5 children in the continent. Thus, the aim of this study is to determine the prevalence of DBM and degree of socioeconomic inequality in double burden of malnutrition among children under 5 years in sub-Saharan Africa.”
Comment 5 Reviewer 1
Line 79 - should this be 'comprised brain development' rather than just 'brain development'?.
Response to Comment 5 from Reviewer 1
Thank you this has been resolved, the suggested changes have been incorporated in line 57.
Comment 6 Reviewer 1
Line 147-148 - should these be 'under 60 months' rather than 'under 59 months'.
Response to Comment 6 from Reviewer 1
Thank you this has been resolved, the suggested changes have been incorporated in line 131.

Reviewer 2 Report
Thank you very much for letting me know the socio-economic inequalities in the double border of malnutrition among underfive children in 10 selected Saharan African countries The writing quality is good, the structure is reasonable, and the logic is clear. Some small problems may help improve the quality of the paper:
1. Abstract: The content is too long. Please reduce the number of words to 250-350. For example, there is no need to list the names of 10 countries, and there is no need to list too many data.
2. Introduction: Can you add 2-5 relevant cases from other regions, such as India, China, Bangladesh and other countries. For reference:
DOI:10.2147/RMHP.S382812.
3. Results: For example, what does "Error! Reference source not found.." mean on P205-216?
4. "5. Study strengths and limitations" does not need to be listed as a separate chapter. It can be combined with 4 or 6.
Finally, please consider whether the difference between rural and urban areas has been considered? After all, Table 1 specifically distinguishes between rural and urban areas.
Author Response
Dear International Journal of Environmental Research Public Health (IJERPH) Editorial Office,
Thank you for the opportunity to resubmit a revised draft of our manuscript titled “Socio-economic inequalities in the double burden of malnutrition among under-five children: evidence from 10 selected sub-Saharan African countries” to International Journal of Environmental Research Public Health. We appreciate the time and effort that you and the reviewers have dedicated to providing valuable feedback on our manuscript. We have reviewed the comments and incorporated changes in the manuscript to reflect suggestions provided by the reviewers. We have highlighted the changes within the manuscript and we have also provided responses to some of the comments. Below are detailed responses to each of your valuable comments.
REVIWER 2 COMMENTS AND RESPONSES
Comment 1 Reviewer 2
Thank you very much for letting me know the socio-economic inequalities in the double border of malnutrition among under five children in 10 selected Saharan African countries The writing quality is good, the structure is reasonable, and the logic is clear. Some small problems may help improve the quality of the paper:
Response to Comment 1 from Reviewer 2
We are pleased with the above comment and we thank you for your input which has been relevant to improving the quality of our manuscript.
Comment 2 Reviewer 2
Abstract: The content is too long. Please reduce the number of words to 250-350. For example, there is no need to list the names of 10 countries, and there is no need to list too many data.
Response to Comment 2 from Reviewer 2
We are pleased with the above comment and we thank you for your input. We have since revised the abstract as follows [line 27 to line 45];
“Abstract:
Background: Africa is unlikely to end hunger and all forms of malnutrition by 2030 due to public health problems such as the double burden of malnutrition (DBM). Thus, the aim of this study is to determine the prevalence of DBM and degree of socioeconomic inequality in double burden of malnutrition among children under 5 years in sub-Saharan Africa.
Methods: This study used multi-country data collected by the Demographic and Health Surveys (DHS) Program. Data for this analysis was drawn from the DHS women's questionnaire focusing on children under 5 years. The outcome variable for this study was the Double Burden of Malnutrition (DBM). This variable was computed from 4 indicators: stunting, wasting, underweight and overweight. Inequalities in DBM among children under 5 years were measured using concentration Indices (CI).
Results: The total number of children included in this analysis was 55,285. DBM was highest in Burundi (26.74%), and lowest in Senegal (8.80%). The computed adjusted Erreygers Concentration Indices showed pro-poor socio-economic child health inequalities relative to the double burden of malnutrition. The DBM pro-poor inequalities were more intense in Zimbabwe (-0,0294) and least intense in Burundi (-0,2206).
Conclusions: This study has shown that across SSA, among under-five children, the poor suffer more from the DBM relative to the wealthy. If we are not to leave any child behind, we must address these socio-economic inequalities in sub-Saharan Africa.”
Comment 3 Reviewer 2
Introduction: Can you add 2-5 relevant cases from other regions, such as India, China, Bangladesh and other countries. For reference: DOI:10.2147/RMHP.S382812.
Response to Comment 3 from Reviewer 2
We are pleased with the above comment and we thank you for your input. We have since revised and added a few sentences in the introduction [line 67 to line 73];
“In China, while the prevalence of undernutrition among children has reduce [8], a worrisome new trend is emerging; the co-existence of under-nutrition and over-nutrition [9], and the rising prevalence of overweight and obesity especially in urban areas and affluent rural areas [10]. Despite experiencing economic growth in recent times, India still reports high rates of child malnutrition with 35%, 33% and 17% of children under 5 years being stunted, underweight and wasted, respectively [11].”
Comment 4 Reviewer 2
Results: For example, what does "Error! Reference source not found.." mean on P205-216?
Response to Comment 4 from Reviewer 2
We are pleased with the above comment and we thank you for your input. We were using cross reference when inserting our table and figure citations. This has since been rectified and all tables and figures updated.
Comment 5 Reviewer 2
"5. Study strengths and limitations" does not need to be listed as a separate chapter. It can be combined with 4 or 6.
Response to Comment 5 from Reviewer 2
We are pleased with the above comment and we thank you for your input. We have since incorporated the suggested changes [Line 363 to Line 369].
Comment 6 Reviewer 2
Finally, please consider whether the difference between rural and urban areas has been considered? After all, Table 1 specifically distinguishes between rural and urban areas.
Response to Comment 6 from Reviewer 2
We are pleased with the above comment and we thank you for your input. We have since incorporated the suggested changes and done an urban rural comparison see Table 2 [Line 215 to Line 219].
Across all countries, higher prevalence of DBM were reported in rural areas compared to urban areas [Table 2].
Table 2. Estimated prevalence of children who experience DBM by Residence Status.
|
Country |
Rural % |
SE |
CI |
Urban % |
SE |
CI |
|
Burundi |
29.32 |
0.0064 |
[0.28 0.31] |
12.92 |
0.0108 |
[0.11 0.15] |
|
Ethiopia |
18.85 |
0.0061 |
[0.18 0.20] |
10.12 |
0.0087 |
[0.09 0.12] |
|
Guinea |
22.82 |
0.0082 |
[0.21 0.25] |
16.32 |
0.0116 |
[0.14 0.19] |
|
Malawi |
15.52 |
0.0054 |
[0.15 0.17] |
11.95 |
0.0109 |
[0.10 0.14] |
|
Mali |
15.44 |
0.0045 |
[0.16 0.17] |
12.81 |
0.0070 |
[0.12 0.14] |
|
Senegal |
9.51 |
0.0046 |
[0.09 0.11] |
7.07 |
0.0063 |
[0.06 0.08] |
|
Sierra Leone |
15.80 |
0.0064 |
[0.15 0.17] |
14.14 |
0.0095 |
[0.12 0.16] |
|
South Africa |
18.45 |
0.0146 |
[0.16 0.22] |
14.13 |
0.0127 |
[0.12 0.17] |
|
Zambia |
17.07 |
0.0047 |
[0.16 0.18] |
16.11 |
0.0071 |
[0.15 0.18] |
|
Zimbabwe |
14.08 |
0.0060 |
[0.13 0.15] |
13.02 |
0.0076 |
[0.12 0.15] |

Round 2
Reviewer 1 Report
I have checked through the manuscript. All points raised have been addressed. I have no further questions about the manuscript. This is an interesting arrticle pointing out the relationship between DBM and socioenconomic inequalities. I'm sure many researchers and policy-makers will find this research to be valuable.